# Tag-Array-Based UHF Passive RFID Tag Attitude Identification of Tracking Methods

**DOI:** 10.3390/s24196305

**Published:** 2024-09-29

**Authors:** Honggang Wang, Sicheng Li, Yurun Zhou, Yongli Wang, Ruoyu Pan, Shengli Pang

**Affiliations:** College of Communication and Information Engineering, Xi’an University of Posts and Telecommunications, Xi’an 710121, China; lisicheng@stu.xupt.edu.cn (S.L.); zhouyurun@stu.xupt.edu.cn (Y.Z.); wangyongli@stu.xupt.edu.cn (Y.W.); panruoyu@xupt.edu.cn (R.P.); pangshengli@xupt.edu.cn (S.P.)

**Keywords:** UHF RFID, tag array, rotation angle, coupled model, attitude identification

## Abstract

Attitude information is as important as position information in describing and localizing objects. Based on this, this paper proposes a method for object attitude sensing utilizing ultra-high frequency passive RFID technology. This method adopts a double tag array strategy, which effectively enhances the spatial freedom and eliminates phase ambiguity by leveraging the phase difference information between the two tags. Additionally, we delve into the issue of the phase shift caused by coupling interference between the two tags. To effectively compensate for this coupling effect, a series of experiments were conducted to thoroughly examine the specific impact of coupling effects between tags, and based on these findings, a coupling model between tags was established. This model was then integrated into the original phase model to correct for the effects of phase shift, significantly improving the sensing accuracy. Furthermore, we considered the influence of the object rotation angle on phase changes to construct an accurate object attitude recognition and tracking model. To reduce random errors during phase measurement, we employed a polynomial regression method to fit the measured tag phase information, further enhancing the precision of the sensing model. Compared to traditional positioning modes, the dual-tag array strategy essentially increases the number of virtual antennas available for positioning, providing the system with more refined directional discrimination capabilities. The experimental results demonstrated that incorporating the effects of inter-tag coupling interference and rotation angle into the phase model significantly improved the recognition accuracy for both object localization and attitude angle determination. Specifically, the average error of object positioning was reduced to 12.3 cm, while the average error of attitude angle recognition was reduced to 8.28°, making the method suitable for various practical application scenarios requiring attitude recognition.

## 1. Introduction

In recent years, ultra-high frequency radio frequency identification (UHF RFID) technology, as one of the core technologies of the Internet of Things (IoT), has been widely adopted in various industries due to its low cost and passive perception capabilities. This technology effectively enhances logistics efficiency and optimizes supply chain management [1,2,3]. Simultaneously, through non-contact bidirectional communication between the reader and the tag, UHF RFID provides robust support for precise object and personnel tracking and positioning technology. By utilizing the characteristic information returned by the tag, it can achieve object or personnel positioning. Compared to traditional global positioning system (GPS) or other positioning technologies, UHF RFID offers advantages such as ease of deployment, low power consumption, and low cost. Its broad range of applications includes indoor navigation, logistics and warehouse management, personnel tracking, library management, and other fields [4,5,6,7,8].

Meanwhile, along with the in-depth evolution of IoT technology and the concept of Industry 5.0, more and more scenarios have an increased demand for object attitude recognition, such as automotive production part assembly, merchandise tagging, and painting. In this type of production line, it is crucial to grasp the attitude information of the objects. Accurate spatial attitude information can help robots or automation systems accurately assemble parts in the correct position and direction, thus ensuring the quality and efficient assembly of the products. At this stage, however, obtaining this information often requires the use of sophisticated sensors that perform well in tracking large objects such as aircraft or missiles, but have limitations in tracking low-value or small-sized objects. For example, tagging and painting operations for low-cost objects are impractical using high-precision sensors, so a low-cost method is needed to sense the attitude of objects in such scenarios.

In order to solve the above problems, this paper proposes an object attitude sensing method based on UHF passive RFID technology, combined with the object rotation in space, with the following contributions: innovatively adopting a dual-tag array method, simultaneously enhancing spatial freedom, using the phase information between the two tags to eliminate phase ambiguity, considering the phase shift caused by the coupling interference between the two tags, and evaluating the tag coupling effect through several experimental tests. The phase compensation after tag coupling was determined through several experimental tests, so as to realize accurate object attitude sensing, relying on the characteristics of UHF RFID technology itself. The system does not need an additional power supply. Through the design of an object attitude recognition tracking algorithm, the system can capture and analyze an object’s motion and positional changes in a timely manner, thus giving the solution a wide range of applicability and flexibility in diverse application scenarios. This series of technological innovations provides an efficient and feasible solution for accurately tracking objects of different sizes and values.

The rest of the paper is organized as follows: Section 2 presents related work. Section 3 describes the system architecture, as well as detailing the construction of the spatial attitude perception model. Section 4 describes the experimental scenarios and details of the experimental validation. Section 5 summarizes the methodology.

## 2. Related Works

The UHF RFID system has a wide range of potential applications in spatial location tracking due to its unique backscatter communication principle. Since the RSSI (received signal strength indication) and phase information of RF signals are easy to obtain, most of the existing research work focuses on positioning and tracking applications based on RSSI and phase information [9,10,11,12,13,14,15,16,17,18,19,20,21,22,23,24,25,26,27,28] . However, RSSI-based schemes are susceptible to multipath effects and have limited positioning accuracy. Therefore, using the phase information of the signal for positioning and tracking has become the main sensing pathway, and related works on phase sensing will be highlighted next.

With the increasing demand for target localization, researchers have not only pursued higher positioning accuracy, but have also started to explore the attitude information of tags in space. Tagoram [11] solved the intrinsic phase deviation of individual tags using a form of phase differencing, and designed a high-accuracy position estimation method based on differential holograms. Wang [12] achieved higher accuracy in target positioning by deploying multiple RFID antennas and using multi-antenna phase combinations to simulate beamforming to locate the target, thus achieving higher accuracy. In addition, refs. [13,14] pointed out that the phase of a tag is not only determined by the distance between it and the reader antenna, but also affected by its relative attitude angle to the reader antenna. Tagyro [13] achieved gyroscope-like rotation angle sensing by attaching a set of passive RFID tags to act as orientation sensors for everyday objects, while RF-Prism [15] used phase information for a multifunctional sensing approach, solving the phase entanglement problem through an integrated model and a complete design to achieve object localization and orientation tracking. OmniTrack [16] constructed an orientation-aware phase model to achieve accurate sensing of the target position and rotational direction by quantifying the effect of tag distance and tag orientation on phase changes. Lin [17] modified the tag by processing the phase difference between the transmitted and received signal responses, detecting the displacement and tilt angle of the tag attached to the target object according to the change in the phase information of the tag response.

Trajectory tracking focuses on the continuous change in target location points over a period of time, which can be regarded as the expansion of positioning technology into time continuity, thus forming a complete motion trajectory. However, moving objects increase the system positioning error, and in order to improve the positioning accuracy, spatial awareness, and tracking performance, researchers have to consider increasing the number of antennas or the number of antennas [18,19,20,21,22,23,24]. Tagball [18] used four antennas to track the positional change of an object with multiple tags attached to it in space; Zeng [19] constructed a virtual antenna array by collecting phases from mobile antennas, and further used the mobile antennas to construct multiple hyperbolas for localization; BackPos [22] deployed multiple co-axial reader antennas on one side of the localization area, and constructed a hyperbola using the phase difference to locate and track the target. In addition, the use of tag arrays [20,21,24] not only improves the tracking accuracy, but also solves the phase blurring problem when the object is moving [19,21,22], but this requires the tag array spacing to be controlled within half of the half-wavelength. Tag- Compass [23] utilized the polarization characteristics of RF waves for orientation measurement, and through signal processing and deviation minimization steps, inferred the orientation at which the tag was attached; RF-Dial [24] continuously tracked objects in translation and rotation by attaching a set of tags to the surface of the object, in combination with an orthogonal antenna; PSOTrack [25] proposed a dynamic trajectory tracking algorithm, which achieved effective tracking of the object by means of a preprocessing module and an optimized particle swarm algorithm; the studies in [14,18,24,26] further extended 2D trajectory tracking to provide integrated sensing of the displacement and rotation of the target in space. The works in [27,28] monitored the target tag using at least two antennas with known positions, and estimated the tag’s trajectory by collecting and analyzing the phase information measured by these antennas.

In summary, this paper proposes an object attitude recognition tracking method based on UHF passive RFID technology. Combined with the rotation of the object in space, a dual-tag array is used to achieve accurate object attitude sensing.

## 3. System Architecture

This system uses a dual-antenna and dual-tag array deployment method for object attitude recognition tracking, and this chapter details the signal phase used. The influencing factors that cause the phase information to be shifted are analyzed, i.e., the coupling effect between the tag arrays and the influence of the rotation angle, and a mathematical model is further developed to take into account the factors affecting the phase information, phase signal model reconstruction is carried out, and finally a tag -attitude-aware tracking model is established.

### 3.1. RFID Phase Signal Model

Traditional RSSI measurements cannot provide highly accurate positioning and tracking information, especially in complex multipath environments where the accuracy is even more limited. Phase measurements can compensate for these limitations by providing higher positioning accuracy, stronger anti-jamming capability, and distance measurement capability than RSSI, and therefore have important application value in RFID systems. Existing CTOS readers (e.g., Impinj Speedway R420) can provide phase information with high accuracy, but at the same time introduce a phase ambiguity of π radians, i.e., the output phase may be the true phase value plus/minus π radians, which is a phenomenon that needs to be taken into account and dealt with in the system design.

Figure 1 illustrates a typical signal propagation process between a passive tag and a reader. Variations in phase φ are generated during forward and tag backscatter communication [29], covering the spatial distance between the reader antenna and the tag, as well as the effects of hardware circuitry, such as differences in the reader’s transmitter/receiver circuitry and the tag itself. The basic phase distance model can be expressed as
(1)$$\left\{\begin{array}{c} \varphi =\left(2\pi \frac{2d}{\lambda }+\mu \right)\;\mathrm{mod}\;2\pi \\ \mu =\theta_{T}+\theta_{R}+\theta_{TAG} \end{array}\right.$$
where *d* is the distance between the reader antenna and the tag; λ is the wavelength of the signal; μ represents the phase variation due to the various hardware characteristics, including the reader’s transmitter $\theta_{T}$, the tag’s hardware characteristics $\theta_{TAG}$, and the reader’s receiver $\theta_{R}$. $\theta_{T}$, $\theta_{R}$ is a constant associated with the hardware circuitry, and $\theta_{TAG}$ is usually considered a constant or random noise. However, when the tags face the antenna at different angles or when there are multiple tags in the reader’s read area, the phase change due to the tag rotation angle and the effect of inter-tag coupling effects may make a localization or tracking method based on this phase model inaccurate. Therefore, the following work takes these factors into account within the phase model, for an improvement in phase information accuracy.

### 3.2. Double Tag Array

Figure 2 illustrates the tag array used in this system, where object pose recognition and tracking is achieved by attaching multiple tags to the target object. Although a multi-tag scheme increases the possibility of target pose recognition, a layout with too close proximity between tags may trigger coupling effects, which may affect the received phase measurements and reduce the accuracy of pose recognition and tracking. To address these challenges, this section delves into strategies to mitigate the coupling effect and improve the sensing and localization accuracy.

### 3.3. Error Analysis and Modeling

In addition to the hardware factors of the reader and the tag itself, the distance and angle of the tag in space relative to the reader antenna, as well as the coupling between the tag arrays, also have an effect on the phase between the reader and the tag in the system. However, how the rotation angle of an object affects the phase change in backscattered signals is not clear, and experimental measurements and analyses were carried out for a better understanding and modeling of tag sensing.

#### 3.3.1. Effect of Tag Rotation Angle on Phase Measurement

As shown in Figure 3, the *x*-axis was defined as a line perpendicular to the antenna plane, and the x−y plane was parallel to the ground. The *z*-axis was perpendicular to the x−y plane, which was rotated around the *x*, *y*, and *z*-axes, respectively, in this scenario, and the phase changes were measured.

The measurement results are shown in Figure 4, the phase change of the tag rotating around the *x*-axis shows a linear characteristic, with the slope of the change exhibiting 2. When the tag rotated around the *y*-axis, there was no significant change in the phase. However, when the tag rotated around the *z*-axis, there were some blind spots between the reader and tag, due to a mismatch in polarization directions. Therefore, during the actual localization process, consideration should be given to placing the tag array on top or on the side of the object, to ensure more reliable reads.

#### 3.3.2. Effect of Tag Array Coupling on Phase Measurements

When tags are in close proximity to each other, the electromagnetic fields between the tags interfere with each other, triggering a severe tag coupling effect. This effect makes the phase value of each tag deviate greatly compared to the uncoupled case, which ultimately leads to a decrease in attitude recognition and positioning accuracy. The positional relationship between the tag array and the antenna must be considered to address this problem. As shown in Figure 5, our system thoroughly investigated four typical positional relationships between the tag array and the reader antenna: the tag array is parallel to the *x*-axis and passes through the *x*-axis, is parallel but does not pass through the *x*-axis, is perpendicular to the *x*-axis, and forms a certain angle with the *x*-axis.

In real scenario applications, these positional relationships are influenced by three key factors: the spacing d of the tag array determines the coupling strength between the tags; the distance *D* from the tag array to the antenna influences the strength and distribution of the electromagnetic field; and the angle θ of the tag array with respect to the *x*-axis of the antenna further complicates the interaction of the electromagnetic field. Figure 6 shows the different states of changes to a tag array.

#### 3.3.3. Coupled Model Construction and Experimental Verification

Using the position relationship of the tag array in Section 3.3.2, combined with the signal transmission model, a coupling model was analyzed and established. When there are multiple tags in the reading area of the reader, only one tag interacts with the reader during each communication process, and the other tags are in an open circuit state, so no backscatter signal will be generated. Therefore, the communication link signal includes the forward path signal from the reader to the tag and the backscatter signal modulated by the tag after acquiring energy [30]. The signal received by the reader can be expressed as
(2)$$\begin{array}{c} S_{R}\left(t\right)=S_{d}\left(t\right)+S_{i}\left(t\right) \end{array}$$
where $S_{d}\left(t\right)$ refers to the signal returned by the reader that directly reaches the tag, and $S_{i}\left(t\right)$ refers to the signal returned by the reader that passes through tag $T_{i}$ and reaches tag *T*. At this time, since the tags are activated almost simultaneously, the coupling of the tags is included in $S_{i}\left(t\right)$. Assuming that the original signal sent by the reader is $S_{0}\left(t\right)$, then
(3)$$\left\{\begin{array}{c} S_{d}\left(t\right)=S_{0}\left(t\right)h_{A}h_{T}h_{A\to T\to A}\\ S_{i}\left(t\right)=S_{0}\left(t\right)h_{A}h_{T}h_{T_{i}}h_{A\to T\to T_{i}\to A} \end{array}\right.$$

The reader receives the signal $S_{R}\left(t\right)$, which can be further expressed as
(4)$$\begin{array}{c} S_{R}\left(t\right)=S_{0}\left(t\right)h_{A}h_{T}\left(h_{A\to T}+h_{T_{i}}h_{A\to T_{i}\to T}\right)h_{T\to A} \end{array}$$
where $h=\alpha e^{j\theta }$ is a series of complex numbers, called channel parameters, and α and θ represent channel attenuation and its phase offset. Since channel attenuation is not considered, α is set uniformly as a constant *c*, so $h_{A}=ce^{j\theta_{R}}$ represents the signal characteristics of the transmitted signal, and the phase shift $\theta_{R}$ is related to the operating frequency of the reader. $h_{T}$ and $h_{T_{i}}$ represent tag characteristics, respectively, $h_{T}=ce^{j\theta_{T}}$, $h_{T_{i}}=ce^{j\theta_{T_{i}}}$, $\theta_{T}$, and $\theta_{T_{i}}$ represent the phase shift generated by the tag; and $h_{A\to T}$, $h_{T\to A}$, and $h_{A\to T_{i}\to T}$ represent the signal characteristics reflected along the direct path and through the other adjacent tags, respectively.
(5)$$\begin{array}{c} \left\{\begin{array}{c} h_{A\to T}=h_{T\to A}=ce^{j(\frac{2\pi d_{T}}{\lambda })}\\ h_{A\to T_{i}\to T}=ce^{j(\frac{2\pi D_{T}}{\lambda })} \end{array}\right. \end{array}$$
where $d_{T}$ is the distance from the tag *T* to the reader, and $D_{T}=d_{T_{i}}+d_{T_{i}T}$ denotes the sum of the distance between the tag $T_{i}$ to the reader antenna $d_{T_{i}}$ and the distance between the two tags $d_{T_{i}T}$. At this point, by using the channel transmission characteristics, the channel characteristics of the inter-tag coupling effect can be further solved as
(6)$$\begin{array}{c} h_{T_{i}T}=\frac{h_{A\to T_{i}\to T}}{h_{A\to T}}=ce^{j\left(\frac{2\pi D_{T}-d_{T}}{\lambda }\right)} \end{array}$$

The channel characteristics of the received signal $S_{R}\left(t\right)$ can thus be expressed as
(7)$$\begin{array}{c} H=h_{A}h_{T}h_{A\to T}\left(1+h_{T_{i}}h_{T_{i}T}\right)h_{T\to A} \end{array}$$
extracting phase information in the received baseband signal, comprising a tag-coupled phase signal
(8)$$\begin{array}{c} \varphi =Arg\left(H\right)=\varphi_{T}+\varphi_{c}+\theta_{R}+\theta_{T_{i}}+\theta_{T} \end{array}$$
where $\mathrm{Arg}\left(\cdot \right)$ is the mathematical function used to return the spoke angle of the complex number, $\varphi_{T}=4\pi d_{T}/\lambda$ refers to the signaling of the tags in the direct path, $\varphi_{c}=2\pi \left(D_{T}-d_{T}\right)/\lambda$ is the coupling term between the tags, and $\theta_{R}$, $\theta_{T_{i}}$, $\theta_{T}$ and have already been presented in the previous section and will not be explained here.

Scenario (a). The test situation is shown in Figure 7. When the communication link distance exceeds one wavelength, there will be a phase ambiguity problem. To eliminate phase ambiguity, the distance difference for the communication link between the antenna and adjacent tags should be less than 1/2 wavelength, i.e., $\left|d_{T_{i}}-d_{T_{i+1}}\right|<\lambda /2$ . Due to the round-trip distance between the reader and the tag, the distance between the tags should satisfy d<λ/4. To meet the above situation of tag spacing (about 8 cm), the experiment set the range of tag spacing variation to 3–8 cm, and tested situation (a) within this range to see the optimal spacing d of the tag array within this spacing.

The experiment set the tag array to a distance of 100 cm from the antenna, and collected the tag spacing between 3 cm and 8 cm with a collection interval of 1 cm. Multiple tests were conducted. The phase shift of the tag increased from 0.0506 rad at 3 cm to 0.7268 rad at 8 cm, indicating that the tag spacing between 3 cm and 6 cm was more appropriate. Therefore, in this article, the tag spacing *d* was set to 5 cm. In addition, according to the coupling model, if the tag spacing is much smaller than the distance from the tag array to the antenna, the influence of tag spacing and tag to antenna distance on the received phase coupling term can be ignored.

Scenario (b). As shown in Figure 8a,b, the tag array was moved horizontally along the *x*-axis, either parallel or perpendicular to the *x*-axis. As the distance *D* from the tag array to the antenna changed, a single tag was first tested, and then the same operation was performed using the tag array to obtain the phase change of a single tag without the coupling effect and with coupling.

The experimental test results showed that, combined with the coupling model, if the distance $d_{T_{i}T}$ between tags is far less than the distances $d_{T}$ and $d_{T_{i}}$ from the antenna to the dual tags, then the value of $D_{T}-d_{T}$ in the phase coupling term can be ignored. Therefore, it can be concluded that the phase coupling term between tags is basically independent of the positional relationship *D* of the tags relative to the reader antenna.

Based on the experimental results from Section 3.3.1, which showed a linear pattern in phase when the tag was rotated around the *x*-axis, and considering the importance of this linear pattern in phase information, this experiment focused on coupling tests of the tag array under *x*-axis rotation conditions. Having already verified the phase change characteristics of a single tag relative to the antenna polarization direction, when extending this test to the tag array, it was observed that the phase changes between the two adjacent tags still followed the pattern described in Section 3.3.1. In Figure 9, the phase change pattern of the tag array during rotation is displayed. Specifically, the phase shift was first tested with tag $T_{1}$ rotating around the *x*-axis, followed by adding tag $T_{2}$ and performing the same test. The phase offset is illustrated in Figure 8c. The experimental results suggested that the rotation angle of the array did not significantly affect the phase, indicating that, in practical applications, the coupling effect of the tag array relative to the antenna angle does not require excessive attention.

### 3.4. Decomposition of Motion of Objects

During the continuous motion of an object, its position and attitude are constantly changing; that is, the continuous motion performed by the object in a plane covers translation and rotation. In this section, the tag rotation law is taken into account in the original phase model, how to solve the phase ambiguity problem using double tags to achieve localization is described, and finally the construction of an orientation-aware model is performed for the process of the object’s motion.

#### 3.4.1. Phase Model Reconstruction

In the empirical study in Section 3.3, when the circularly polarized antenna read a rotating tag, its phase was linearly transformed with the change in angle at a rate where *k* was ±2; in addition, the coupling effect between the tag arrays also had an effect on the phase, and when the rotational term and the coupling term were introduced, the RF phase received by the RFID reading antenna could be expressed as
(9)$$\begin{array}{c} \left\{\begin{array}{c} \varphi =\left(2\pi \cdot \frac{2d}{\lambda }+k\theta +\varphi_{c}+\mu \right)\;\mathrm{mod}\;2\pi \\ \mu =\theta_{R}+\theta_{T}+\theta_{TAG} \end{array}\right. \end{array}$$
where the newly introduced variable θ represents the phase change caused by the tag rotation direction, and $\varphi_{c}$ is the tag coupling term. This phase eigenvalue will be used as the basis for subsequent calculation. By attaching a group of tag arrays to the object surface, the attitude change of the target object can be calculated according to the established model.

#### 3.4.2. Hyperbolic Construction

Hyperbolic construction relies on the virtual antenna working in concert with the physical antenna. The virtual antennas were first accurately constructed based on the geometrical positions of the tag arrays, i.e., the arrangement, the relative distances between the tags, and the angular relationship between them and the physical antennas. Due to the existence of the dual-tag array, each physical antenna was functionally considered to coexist with a set of virtual antennas that were parallel to the tag array, and the angles between them were kept consistent with those in the tag array.

The process of constructing a virtual antenna is shown in Figure 10. At this point, the antenna and the virtual antenna can be set as two foci. Assuming that the coordinates of antenna $AT_{1}$ are $\left(x_{1},y_{1}\right)$, and the initial angle between the two tags, the coordinates of the virtual antenna $vAT_{1}$ are $\left(x_{v1},y_{v1}\right)=\left(x_{1}-d\cos\theta ,y_{1}-d\sin\theta \right)$.  Therefore, $AT_{1}$ and $vAT_{1}$ are the two focal points of the hyperbola, and the center points of the two focal points are
(10)$$\begin{array}{c} \left(x_{o},y_{o}\right)=\left(\frac{x_{1}+x_{v1}}{2}\frac{y_{1}+y_{v1}}{2}\right)=\left(\frac{2x_{1}-dcos\theta }{2}\frac{2y_{1}-dsin\theta }{2}\right) \end{array}$$

Since the two focus positions are the coordinates after rotation, the original coordinates should also be rotated. Using the rotation formula, the original coordinate point can be obtained after the rotation of the coordinates, and the corresponding relationship is
(11)$$\begin{array}{c} \left[\begin{array}{c} x^{\prime }\\ y^{\prime } \end{array}\right]=\left[\begin{array}{cc} \begin{array}{c} \cos\theta \\ -\sin\theta \end{array} & \begin{array}{c} \sin\theta \\ \cos\theta \end{array} \end{array}\right]\left[\begin{array}{c} x\\ y \end{array}\right] \end{array}$$

The hyperbolic equation is constructed as
(12)$$\begin{array}{c} \frac{{\left(\left(x-x_{o}\right)\cos\theta +\left(y-y_{o}\right)\sin\theta \right)}^{2}}{a^{2}}-{\frac{\left(\left(x-x_{o}\right)\sin\theta +\left(y-y_{o}\right)\cos\theta \right)}{b^{2}}}^{2} \end{array}$$
where
$$\left\{\begin{array}{cc} a & =\frac{\Delta d}{2}=\frac{\lambda }{8\pi }\left(\Delta \varphi +2k\pi \right)\\ c & =\frac{d}{2}\\ b & =\sqrt{c^{2}-a^{2}} \end{array}\right.$$

To achieve the localization process, at least two or more hyperbolas are required to find the intersection point, and if more than one antenna is deployed in the scene, each antenna corresponds to one hyperbola, so at least two antennas are required to achieve localization in this system. There is also the problem that the value of *a* is not unique, and phase blur removal is required.

#### 3.4.3. Phase Deblurring

Based on the above content, if there are target tags $T_{1}$ and $T_{2}$ in the scene, and $d(AT_{1},T_{k})$ is the distance from antenna $AT_{1}$ to tag $T_{k}$, then the antenna can measure the phase of the two tags as
(13)$$\begin{array}{c} \left\{\begin{array}{c} \varphi_{1}=\left(2\pi \cdot \frac{2d(AT_{1},T_{1})}{\lambda }+\theta_{R}+\theta_{T}+\theta_{TAG1}\right)\;\mathrm{mod}\;2\pi \\ \varphi_{2}=\left(2\pi \cdot \frac{2d(AT_{1},T_{2})}{\lambda }+\theta_{R}+\theta_{T}+\theta_{TAG2}\right)\;\mathrm{mod}\;2\pi \end{array}\right. \end{array}$$
where $\theta_{e}=\theta_{R}+\theta_{T}$. The difference in distance from the antenna to the two tags can be obtained by subtracting Equation  (Equation 13)
(14)$$\Delta d=\frac{\lambda }{4\pi }\left(\Delta \varphi +2k\pi \right)$$
where $\Delta \varphi =\left|\varphi_{1}-\varphi_{2}\right|$, $\Delta d=\left|d\left(AT_{1},T_{1}\right)-d\left(AT_{1},T_{2}\right)\right|$, Δd are not unique; therefore, the tag spacing *d* is known and according to the triangle constraint, the sum of the lengths of any two sides of the triangle is greater than the third side. When d<λ/4, as shown in Figure 11, and when Δd<d, Equation  (Equation 15) can be used to limit the phase ambiguity, so that a hyperbola of the target object constrained to the curve can be constructed
(15)$$\Delta d=\left\{\begin{array}{cc} \frac{\lambda }{4\pi }\left(\Delta \varphi \right) & \mathrm{if}\;\Delta \varphi \cdot \Delta d>0\\ \frac{\lambda }{4\pi }\left(\Delta \varphi -2\pi \right) & \mathrm{if}\;\Delta \varphi >0\;\mathrm{and}\;\Delta d<0\\ \frac{\lambda }{4\pi }\left(\Delta \varphi +2\pi \right) & \mathrm{if}\;\Delta \varphi <0\;\mathrm{and}\;\Delta d>0 \end{array}\right.$$

#### 3.4.4. Object Motion Tracking Perception Model

Taking the object localization perception methods in Section 3.4.1 and Section 3.4.2, they are now combined for model building, and using a specific analysis of a hypothetical scenario, tag array attitude recognition and tracking model construction is carried out. The tag movement scenario is shown in Figure 12.

According to the above introduction, the phase received by the antenna can be further expressed by introducing a time parameter into Equation (Equation 9) as
(16)$$\begin{array}{c} \left\{\begin{array}{c} \varphi_{p,q,t}=\left(2\pi \cdot \frac{2d_{p,q,t}}{\lambda }+k\theta_{p,q,t}+\varphi_{c,t}+\mu \right)\;\mathrm{mod}\;2\pi \\ \mu =\theta_{R}+\theta_{T}+\theta_{TAG} \end{array}\right. \end{array}$$
where $\varphi_{c,t}=2\pi \cdot \frac{d_{p,1,t}+\Delta d-d_{p,2,t}}{\lambda }$, Δd denote the distance difference between the tags; $\varphi_{p,q,t}$, $d_{p,q,t}$, and $\theta_{p,q,t}$ denote the phase, relative distance, and phase offset due to rotation of the tag *q* received by the antenna *p* at moment *t*, respectively. The system tag attached to the object dynamic movement, through the t+1 moment of phase minus *t* moment phase, where the sampling interval Δt phase difference is $\Delta \varphi_{p,q,t}$, can obtain the tag at a moving distance, and the phase subtraction process eliminates the hardware transmitter circuit and receiver circuit caused by the error. In addition, due to the use of the same tag, the tag production process caused by the phase error is ignored, i.e., set $\theta_{TAG1}=\theta_{TAG2}$, so the tag at different moments of the phase difference is
(17)$$\begin{array}{c} \Delta \varphi_{p,q,t}=2\pi \cdot \frac{2\Delta d_{p,q,t}}{\lambda }+\Delta \varphi_{c,t}+k\Delta \theta_{p,q,t} \end{array}$$
where $\left\{\begin{array}{c} \begin{array}{c} \Delta \varphi_{p,q,t}=\left|\varphi_{p,q,t+1}-\varphi_{p,q,t}\right|\\ \Delta \theta_{p,q,t}=\left|\theta_{p,q,t+1}-\theta_{p,q,t}\right| \end{array}\\ \Delta \varphi_{c,t}=\left|\varphi_{c,t+1}-\varphi_{c,t+1}\right| \end{array}\right.$, since the change in displacement contains the moving distance and the moving direction, so $\Delta d_{p,q,t}=\left|\vec{d_{p,q,t+1}}-\vec{d_{p,q,t}}\right|$. Meanwhile, since the two tags are placed in parallel, their rotation angles relative to the antenna are the same at different moments, so this results in the same rotational phase shift, thus $\Delta \theta_{t}=\Delta \theta_{p,1,t}=\Delta \theta_{p,2,t}$. The coordinates of tag *q* at the next moment are set to be $\left(x_{q,t+1},y_{q,t+1}\right)$, so the displacement of tag *q* within Δt is
(18)$$\begin{array}{c} \Delta d_{p,q,t}=\sqrt{x_{q,t+1}-{x_{t}}^{2}+y_{q,t+1}-{y_{t}}^{2}} \end{array}$$

By reconstructing the change pattern of the phase signal, the total phase change minus the phase of the tag rotation is the actual phase change generated by the tag movement, which is brought into the $\Delta d_{p,q,t}$. Combining with equation Equation (Equation 16), we can obtain
(19)$$\begin{array}{c} \sqrt{x_{q,t+1}-{x_{t}}^{2}+y_{q,t+1}-{y_{t}}^{2}}=\frac{\lambda }{4\pi }\left(\Delta \varphi_{p,q,t}+\Delta \varphi_{c,t}-k\Delta \theta_{t}\right) \end{array}$$

Combined with the construction of the hyperbola in Section 3.4.2, after obtaining multiple hyperbola equations, the solution process is simplified by transforming the set of hyperbola equations, and the label positioning is regarded as an optimization problem, where the transformed objective function is
(20)$$\begin{array}{c} J=\sum_{p}^{n}{\left(d\left(T,A_{p}\right)-d\left(T,A_{p-1}\right)-\Delta d_{p,t}\right)}^{2} \end{array}$$
equals
(21)$$\begin{array}{cc} J= & \sum_{p=1}^{2}\sum_{q=1}^{2}[\sqrt{{\left(x_{q,t+1}-x_{p}+d\cos\Delta \theta_{t}\right)}^{2}+{\left(y_{q,t+1}-y_{p}+d\sin\Delta \theta_{t}\right)}^{2}}\\ \; & -\sqrt{{\left(x_{q,t+1}-x_{p}\right)}^{2}+{\left(y_{q,t+1}-y_{p}\right)}^{2}}-\frac{\lambda }{4\pi }\left(\Delta \varphi_{p,t}+2k\pi \right)] \end{array}$$

After the objective function has been established, the parameter can be solved through an optimization method, but there is still a problem in the formula, that is, the phase difference $\Delta \varphi_{p,t}$ may have a random error. The impact of this error will be further analyzed in Section 3.5.

### 3.5. Stochastic Phase Polynomial Regression

The phase information collected during the movement of the tag was accompanied by some inherent problems such as ambiguity, phase error due to the hardware characteristics, and phase random noise. According to the hyperbolic localization method used in a previous paper, the process eliminates the phase ambiguity problem and the phase error problem caused by hardware errors. For the random error generated by phase noise, according to previous studies [18,31,32] and experimental verification, the noise of the phase signal obeys a Gaussian distribution during the tag acquisition process, and Figure 13a shows the distribution of 107 phase measurements measured by the reader at 920.675 MHz frequency when the tag was fixed at a distance of 100 cm. According to the distribution of the phase, using polynomial regression compensates for the measured phase.

According to Section 3.4, the model representation of the phase value Δφ can be expressed as
(22)$$\begin{array}{c} \Delta \varphi =\Delta \hat{\varphi }\left(t\right)+\zeta \end{array}$$
where $\Delta \hat{\varphi }\left(t\right)=\beta_{0}+\beta_{1}t+\beta_{2}t^{2}+\cdots +\beta_{n}t^{n}$ is the value of the time-dependent phase function; $\beta_{0}$, $\beta_{1}$, $\beta_{2}$, $\beta_{3}$ is the polynomial coefficient; *n* is the order of the polynomial; and ζ represents the phase noise.

To find the optimal polynomial coefficients $\left(\beta_{0},\beta_{1},\beta_{2},\cdots \beta_{n}\right)$, define the mean square error cost function as
(23)$$\begin{array}{c} J\left(\beta \right)=J\left(\beta_{0},\beta_{1},\beta_{2},\cdots \beta_{n}\right)\\ =\frac{1}{2m}\sum_{i=1}^{m}{\left(\Delta \varphi_{i}-\Delta \hat{\varphi }\left(t_{i};\beta_{0},\beta_{1},\beta_{2},\cdots \beta_{n}\right)\right)}^{2} \end{array}$$
where *m* is the number of data points, $\Delta \varphi_{i}$ is the phase value of the *i* data point, and $\hat{f}\left(t_{i};\beta_{0},\beta_{1},\beta_{2},\cdots \beta_{n}\right)$ is the prediction value of the polynomial at $t_{i}$. By minimizing $J\left(\beta_{0},\beta_{1},\beta_{2},\cdots \beta_{n}\right)$, the best polynomial coefficient $\beta_{0},\beta_{1},\beta_{2},\cdots \beta_{n}$ is found to minimize the prediction error of the polynomial in the given dataset. Before using the gradient descent algorithm to solve the cost function, scale the data to ensure that the scaling ratio of each feature is roughly the same, improve the convergence speed, and standardize each timestamp $t_{i}$.
(24)$$\begin{array}{c} t_{i}=\frac{t_{i}-\mu_{t}}{\sigma_{t}} \end{array}$$
where $\mu_{t}=\frac{1}{m}\sum_{i=1}^{m}t_{i},\sigma_{t}=\sqrt{\frac{1}{m}\sum_{i=1}^{m}{\left(t_{i}-\mu_{t}\right)}^{2}}$ . Iterate the gradient descent algorithm with the normalized as the new variable and repeat $\beta_{n}=\beta_{n}-\alpha \frac{\partial J\left(\beta \right)}{\partial \beta_{n}}$ until convergence, where α is the learning rate, which controls the step size of each iteration, and $\frac{\partial J\left(\beta \right)}{\partial \beta_{j}}$ is the partial derivative of the cost function with respect to $\beta_{j}$, denoted as
(25)$$\begin{array}{c} \frac{\partial J\left(\beta \right)}{\partial \beta_{n}}=\frac{1}{m}\sum_{i=1}^{m}\left(\Delta \varphi_{i}-\Delta \hat{\varphi }\left(t_{i};\beta_{0}\beta_{1}\beta_{2}\cdots \beta_{n}\right)\cdot t_{i}^{n}\right) \end{array}$$

Figure 13b shows a polynomial regression diagram of the phase value measured by the tag movement. Through the above processing, the predicted value after the multiple linear regression is placed into Equation (Equation 21), i.e., the objective function is
(26)$$\begin{array}{c} \begin{array}{c} J=\underset{p=1}{\sum^{2}}\;\underset{q=1}{\sum^{2}}\;\sqrt{{\left(x_{q,t+1}-x_{p}+d\cos\Delta \theta_{t}\right)}^{2}+{\left(y_{q,t+1}-y_{p}+d\sin\Delta \theta_{t}\right)}^{2}}\\ -\sqrt{{\left(x_{q,t+1}-x_{p}\right)}^{2}+{\left(y_{q,t+1}-y_{p}\right)}^{2}}-\frac{\lambda }{4\pi }\left(\Delta \hat{\varphi }\left(t\right)+2k\pi \right) \end{array} \end{array}$$

### 3.6. Objective Function Optimization

Due to the existence of the deflection angle between antennas, the hyperbolic intersection points may not intersect at a point, and the minimum value is not exactly equal to 0. Therefore, a particle swarm optimization algorithm is needed to solve the optimal parameters. The particle swarm optimization (PSO) algorithm is defined as a swarm intelligence optimization technology, which optimizes the objective function by randomly initializing a swarm of particles in the solution space and then iterating to find the optimal solution. The steps of using PSO to optimize the objective function are as follows:Initialize particle swarm: Randomly assign each particle its initial position and velocity, which is limited to the problem space. The initial position of each particle is also the best position of its individual, and the best position in the whole particle swarm is the global best position.Evaluate particle performance: For each particle, use the target function to evaluate the performance of the current position. If the current position is better than the best position performance of the particle, the best position performance of the particle is updated.Update speed and position: Update the speed and position of each particle based on the best individual position, the best global position, the current speed, and two learning factors (individual and social learning factors).Repeat or terminate: Stop the algorithm if the criteria for termination are met (e.g., the maximum number of iterations has been reached or the global best position improvement is less than the threshold); otherwise, repeat steps 2 and 3.

According to Equation (Equation 26), *J* is the target to be optimized, and the $\left(x_{t+1},y_{t+1},\Delta \theta_{t+1}\right)$ that minimizes *J* is the position and rotation angle of the target tag. The tag dynamic tracking perception process is shown in Algorithm 1.
**Algorithm 1** Mobile label orientation perception algorithm  1:**function** MobiletagOrientationeprception($\Phi_{p,q,t}$, $X_{p}$, $Y_{p}$, *t*, *k*, *d*, λ, θ)  2:    **Initialize** $X_{p},Y_{p}$ for p=1,2, t←0, k←2, d,λ,θ  3:    **Calculate** Δt=(t+1)−t, $\Delta \theta_{t}=\theta_{t+1}-\theta$  4:    **while** Δt≠0 **do**  5:        **for** each n∈N **do**  6:           $\Delta \varphi_{p,q,t}\left[n\right]\leftarrow \varphi_{p,q,t+1}\left[n+1\right]-\varphi_{p,q,t}\left[n\right]$  7:           $\Delta d_{p,q,t}\left[0:n\right]\leftarrow \frac{\lambda }{4\pi }\left(\Delta \varphi_{p,q,t}\left[0:n\right]-k\Delta \theta_{t}\left[0:n\right]\right)$  8:           $\Delta \varphi_{p,t}\left[0:n\right]\leftarrow \varphi_{p,1,t}\left[0:n\right]-\varphi_{p,2,t}\left[0:n\right]$  9:           $\Delta d_{p,t}\left[0:n\right]\leftarrow \frac{\lambda }{4\pi }\left(\Delta \varphi_{p,t}\left[0:n\right]\right)$10:           **if** $\mathrm{sign}\left(\Delta \varphi_{p,t}\left[n\right]\right)=\mathrm{sign}\left(\Delta d_{p,t}\left[n\right]\right)$ **then**11:               $\Delta d_{p,t}\left[n\right]\leftarrow \frac{\lambda }{4\pi }\Delta \varphi_{p,t}\left[n\right]$12:           **else if** $\mathrm{sign}(\Delta \varphi_{p,t}\left[n\right])>0$ & & $\mathrm{sign}(\Delta d_{p,t}\left[n\right])<0$ **then**13:               $\Delta d_{p,t}\left[n\right]\leftarrow \frac{\lambda }{4\pi }\left(\Delta \varphi_{p,t}\left[n\right]+2\pi \right)$14:           **else if** $\mathrm{sign}(\Delta \varphi_{p,t}\left[n\right])<0$ & & $\mathrm{sign}(\Delta d_{p,t}\left[n\right])>0$ **then**15:               $\Delta d_{p,t}\left[n\right]\leftarrow \frac{\lambda }{4\pi }\left(\Delta \varphi_{p,t}\left[n\right]-2\pi \right)$16:           **end if**17:        **end for**18:        **Construct an optimization function according to Equation (Equation 25).**19:        $x_{t+1},y_{t+1},\Delta \theta_{t+1}\leftarrow PSO\_optimization$20:              ▹ Using PSO to optimize. The optimal parameter value when the objective function is closest to 0 (J→0)21:        $x_{t}\leftarrow x_{t+1}$        ▹ Update optimal parameters for next iteration22:        $y_{t}\leftarrow y_{t+1}$23:        $\Delta \theta_{t}\leftarrow \Delta \theta_{t+1}$24:    **end while**25:    **return** $X_{t+1},Y_{t+1},\Delta \Theta_{t+1}$26:**end function**

## 4. Experiment and Result Analysis

For the dynamic tag attitude recognition method studied in Section 3, this section used an Impinj R420 reader writer with multiple circularly polarized antennas with a size of 25 × 25 cm and a gain of 9 dBi, with ISO 18000-6C [33] tags to verify its effectiveness.

### 4.1. Experimental Setup

During the experimental validation process, we designed two antennas and placed them in parallel with a 50 cm spacing. This spacing was chosen to ensure there was enough distance between the antennas to reduce mutual interference, while maintaining continuous and effective coverage of the moving tags. This setup allowed us to capture signals from two different angles simultaneously when the tag underwent dynamic displacement and rotation, thus enhancing the accuracy and robustness of the pose recognition. Additionally, based on the analysis of typical application scenarios (such as warehouses and logistics centers) where tags might be attached, we found that goods and pallets are often stacked to a certain height. Positioning the tag at a height of approximately 100 cm from the ground placed it in the upper-middle range of this stacking height, representing most practical application scenarios. This further ensured the effectiveness of the tag recognition. The tag was experimentally mounted on a mobile robot equipped with a rotating turntable. The robot moved in a straight line within an indoor area, causing the tag to experience changes in both displacement and rotation angles at any given moment. Through this process, we successfully validated the system’s recognition accuracy under conditions of dynamic tag displacement and rotation. The specific scenario is illustrated in Figure 14.

The hardware equipment used in the experiment mainly included the following:(1)Impinj Speedway R420 Reader (Impinj Wireless RF Technology (Shanghai) Co., Ltd., Shanghai, China);(2)2 Laird s9028 PCL antennas (Laird Electronic Materials (Shenzhen) Co., Ltd., Shenzhen, China) and AZ-9662 tag array (Shenzhen Aisen IoT Technology Co., Ltd., Shenzhen, China);(3)Water mobile robot (Beijing Yunji Technology Co., Ltd., Beijing, China);

The reader was configured with Max Throughput, Dual target, Session 0, and a 920.625 MHz operating frequency. In addition, ItemTest software was used to read the data and matlab (9.7.0.1190202 (R2019b)) software was used for data statistics, and the experimental parameters were set as shown in Table 1.

### 4.2. Analysis of Results

Based on the above experimental scenarios and the specific parameter configurations, a linear motion was performed at a constant speed of 0.05 m/s to move the tag along a prescribed route of 1.5 m while it remained stationary without rotation, in order to analyze the accuracy of its trajectory measurement. After 20 repetitions of the experiment had been performed, the error was calculated and determined according to the designed algorithm. The error accuracies in the two cases of introducing the tag coupling term and not introducing the tag coupling term were then compared and the experimental results were presented in the form of graphs. The positioning error of the tag moving without rotation is shown in Figure 15.

When the non-rotating tags were localized and tracked, their average localization error was 9.633 cm, and after considering the introduction of the coupling term, the localization accuracy was significantly improved, and the error was reduced to 7.962 cm. Furthermore, the tag attitude information was introduced to make an evaluation, and the robot carried out the movement while rotating the tag arrays, and the abovementioned scenarios were used for the test. The results of the attitude recognition are shown in Figure 16.

When analyzing the positional accuracy of the tags, we observed that, while incorporating tag orientation information as a positioning parameter enhanced the system’s overall awareness of tag states, it also resulted in reduced accuracy in the *x*-*y* axis direction. Specifically, after including orientation information and associated coupling terms, the overall average positioning error of the tag increased to 13.662 cm, while the angular perception error remained at 8.28°. This result indicates that, although integrating orientation information adds an extra dimension of data, it also increases the complexity and uncertainty of positioning. When the tag rotates, the relative position between its antenna and the reader antenna will change, which may lead to a change in the signal propagation path. Specifically, the rotation may make the antenna direction of the tag deviate from the best receiving direction for the reader antenna, resulting in weakening of the signal strength or signal path blocking, thus affecting the positional accuracy.

To accurately assess this change, we considered several factors that might affect the positional accuracy. First, we carefully adjusted the spacing of the tag array to ensure that the tags at different positions could independently and effectively communicate with the reader. Second, we optimized the reader’s operating frequency to match the tag’s response characteristics, reducing the signal attenuation and interference during transmission. Additionally, recognizing the significant impact of the robot’s movement speed on the tag positioning, we designed multiple experiments to test dynamic tag positioning under varying speed conditions.

(1) Tag array spacing

In an in-depth study of the effect of label spacing on the perception results, the experiments systematically set the spacing of the label arrays, starting from 3 cm and spaced at 1 cm intervals up to 7 cm, for exhaustive experimental testing. In Figure 17, the figure visualizes the effect of tag spacing on the positioning error and angular error.

From the experimental results, it can be seen that the positioning error and angular error were both minimized when the tag spacing was equal to 5 cm, a conclusion that coincides with the experimental conclusions in Section 3.3.2, thus further verifying the reasonableness of setting 5 cm as the spacing of the tag array. In addition, it is worth noting that, due to the reader and the tag hardware itself, the phase shift problem could not be completely avoided. Therefore, although the tag spacing of 6 cm and 7 cm satisfied the condition of less than 4/λ, it still could not completely eliminate the phase blurring according to the triangular constraints, which led to a large error in the results. Based on these considerations, it can be concluded that, under the current experimental conditions, setting the spacing of the tag array to 5 cm was the most reasonable and effective choice.

(2) Reader operating frequency

In order to deeply investigate the effect of frequency variables on recognition accuracy, four frequency bands, 920.625 MHz, 921.125 MHz, 922.875 MHz, and 924.375 MHz, were selected for the experiment. The experimental results are shown in Figure 18a,b, where it can be seen that the perception errors remained almost the same at these four different frequencies, with no significant differences. The results show that the change in frequency had a negligible effect on the positioning accuracy, so there is no clear limitation on the frequency when performing attitude sensing in real scenarios.

(3) Movement speed

In further exploring the effect of different moving speeds on the tag accuracy error, the experiments set three moving speeds of 0.05 m/s, 0.1 m/s, and 0.15 m/s, and the perceived accuracy of the tags was meticulously tested in each of these scenarios. Figure 18c,d reveals that there was a clear correlation between the tested tag accuracy and the movement speed. The faster the tag movement speed, the higher the tracking error. This was due to the fact that when the robot moved at a higher speed, the number of samples taken at each timestamp of the trajectory decreased. Therefore, estimating the target’s motion parameters for each segment introduced a larger error. Specifically, according to the analysis of the test data, when the robot moved at a speed of 0.05 m/s, the number of data samples collected was about three times as much as when the speed was 0.15 m/s. The reduction in data samples was the root cause of the increase in error, so there needs to be a certain limitation on the object’s movement speed in a real-world scenario.

## 5. Conclusions

In industrial application scenarios, the attitude information of an object occupies the same important role as the position information. To meet this demand, this paper proposed a tag spatial attitude recognition tracking method based on tag arrays. The working principle, system implementation, and experimental results of the system were described in detail. A mathematical model to quantify the phase coupling of tags was proposed by deeply considering the phase change caused by tag rotation and the effect of inter-tag coupling on the phase. Furthermore, the phase model of the signal was reconstructed so as to improve the accuracy of the phase information. Finally, a hyperbolic positioning algorithm was implemented on the basis of the model to eliminate the phase ambiguity problem, combined with the particle swarm optimization algorithm to improve the recognition accuracy and simplify the solution process. Compared with other tag localization systems, the attitude sensing introduced by this method is able to obtain more dimensional tag information. After a large number of experimental verifications, the tag attitude recognition system constructed based on this method showed ideal performance.

However, despite the ideal performance in the experiments, there are limitations when applying it to broader, scalable commercial applications. Firstly, dense tag deployment and complex physical environments may increase the signal interference, affecting system stability and accuracy. Secondly, the current computational complexity may limit the system’s real-time performance in large-scale scenarios. Additionally, the stability and durability in extreme environments (e.g., high temperatures, high humidity, or strong electromagnetic interference) need further verification and optimization. To address these limitations, we propose the following future research directions: (1) Investigate more advanced signal processing technologies to better suppress interference and improve system robustness; (2) Optimize algorithm structures to reduce computational complexity and enhance real-time processing capability in large-scale scenarios; (3) Strengthen testing and verification of the system in various extreme environments to ensure stable and reliable operation across different industrial settings. Through these efforts, we aim to continuously refine and expand the method’s application range, providing more efficient and accurate object orientation recognition and tracking solutions for industry.

## Figures and Tables

**Figure 1 sensors-24-06305-f001:**
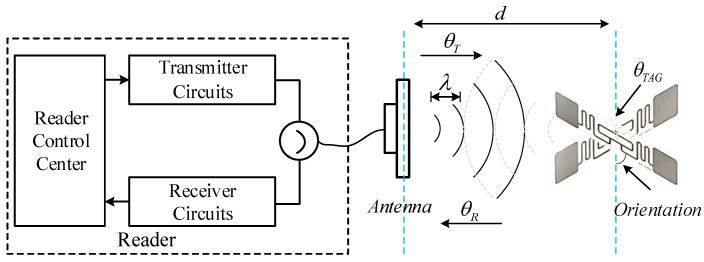
UHF RFID propagation model: detecting the phase change in the radio frequency signal as it propagates between the tag and the reader, and acquiring the phase information of the tag.

**Figure 2 sensors-24-06305-f002:**
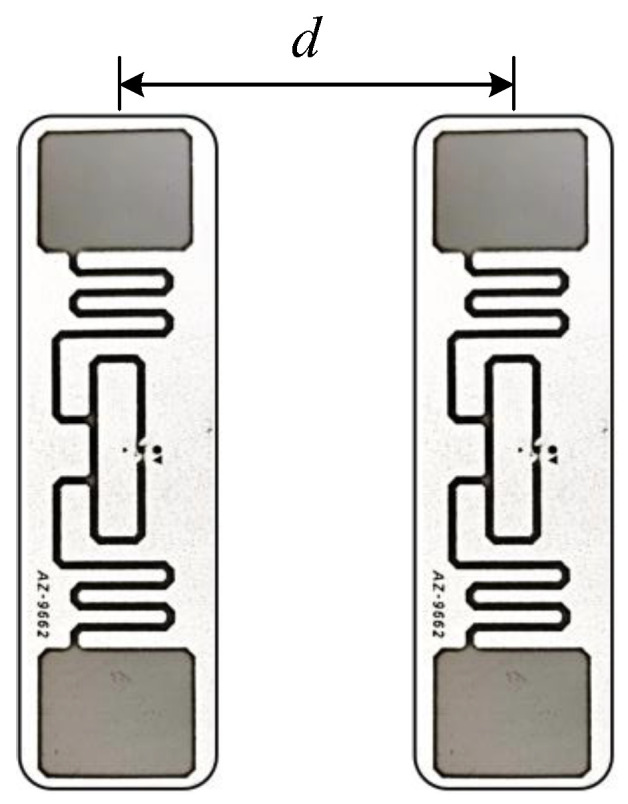
Double Tag Array.

**Figure 3 sensors-24-06305-f003:**
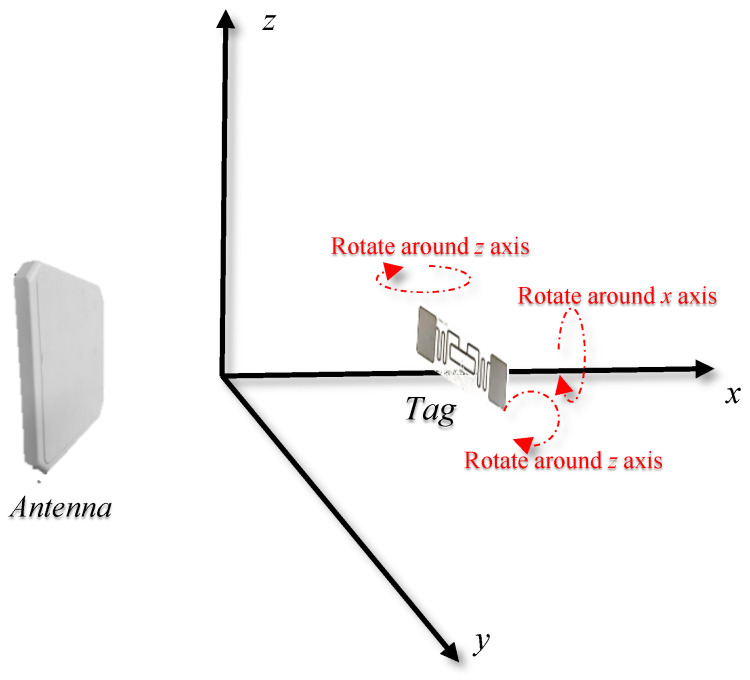
Angle change of the tag relative to the antenna.

**Figure 4 sensors-24-06305-f004:**
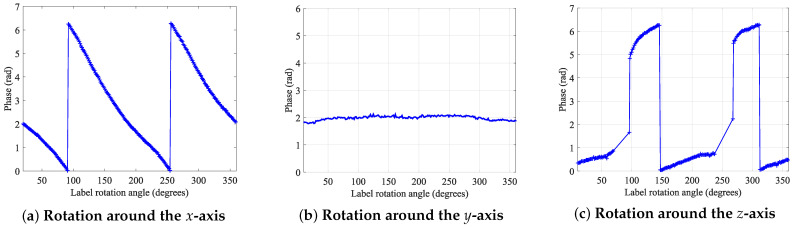
Phase transition caused by tag rotation: The labels were rotated around the *x*-axis, *y*-axis, and *z*-axis of the set coordinates in the changed scene.

**Figure 5 sensors-24-06305-f005:**
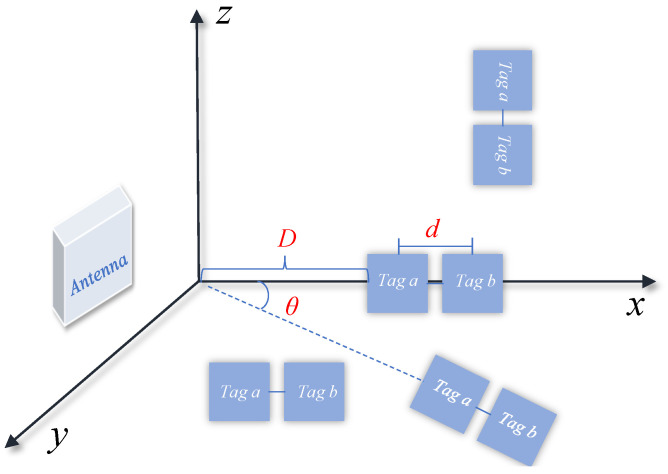
The tag array’s positional relationships with the reader antennae encompass being either parallel to and passing through the *x*-axis, parallel to the *x*-axis but not passing through it, perpendicular to the *x*-axis, or angled with the *x*-axis.

**Figure 6 sensors-24-06305-f006:**
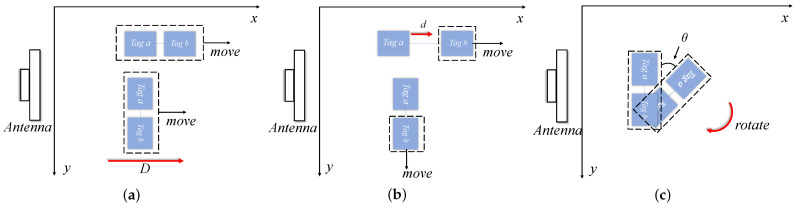
Different states of tag array changes: (**a**) Tag array spacing *d*; (**b**) Distance from tag array to antenna *D*; (**c**) Angle of the tag array relative to the antenna θ.

**Figure 7 sensors-24-06305-f007:**
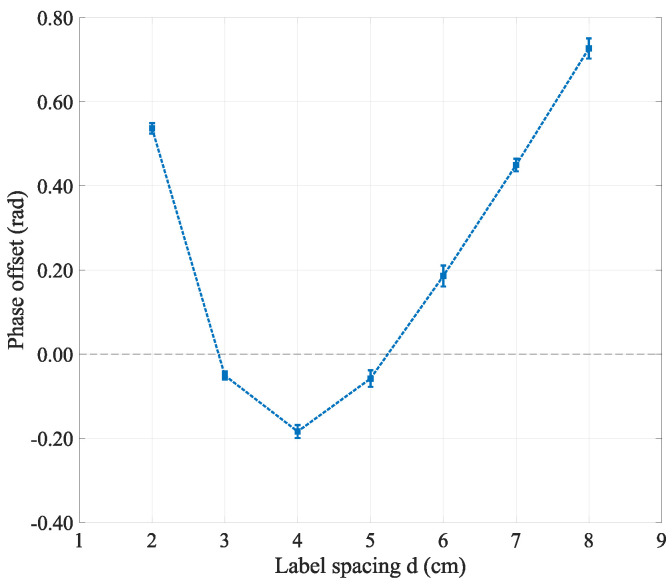
Scenario (a) Test situation: mean and variance of phase bias caused by tag spacing.

**Figure 8 sensors-24-06305-f008:**
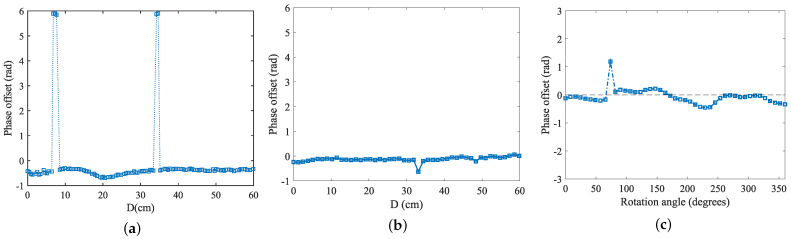
Phase offsets for different situations: (**a**) tag array moves parallel to the *x*-axis; (**b**) tag array moves perpendicular to the *x*-axis; (**c**) tag array rotation.

**Figure 9 sensors-24-06305-f009:**
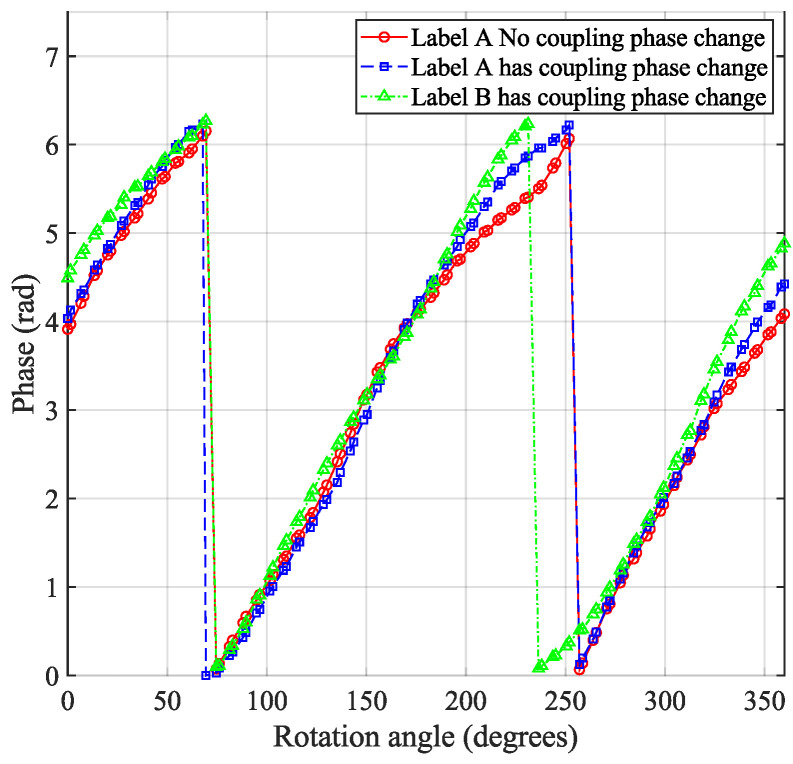
Label array phase changes with rotation angle: when extending from single tag rotation to tag array rotation, the phase exhibited the same variation trend.

**Figure 10 sensors-24-06305-f010:**
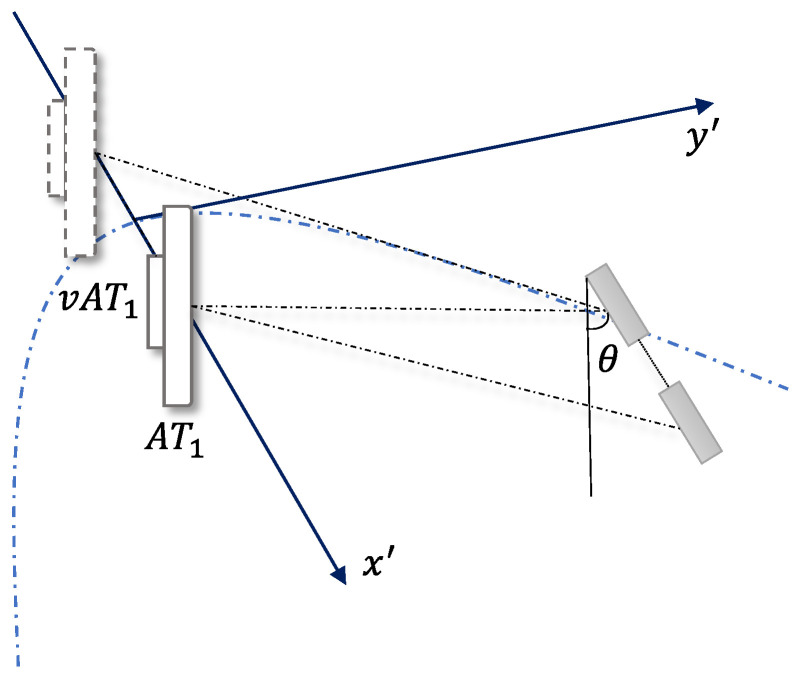
The physical antenna $AT_{1}$ can be regarded as adding a virtual antenna $vAT_{1}$ parallel to the array due to the presence of a two-tag array, and if the angle between the two tags is θ, the angle between the antennas is also kept as θ.

**Figure 11 sensors-24-06305-f011:**
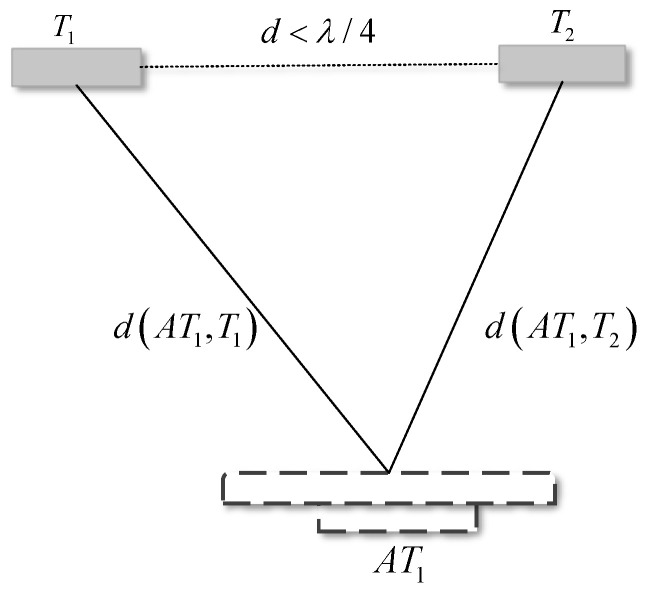
Triangle constraint. When the tag spacing is less than one quarter of the wavelength, the phase difference of the antenna receiving tag array can be controlled within $\left(-\pi ,\pi \right)$.

**Figure 12 sensors-24-06305-f012:**
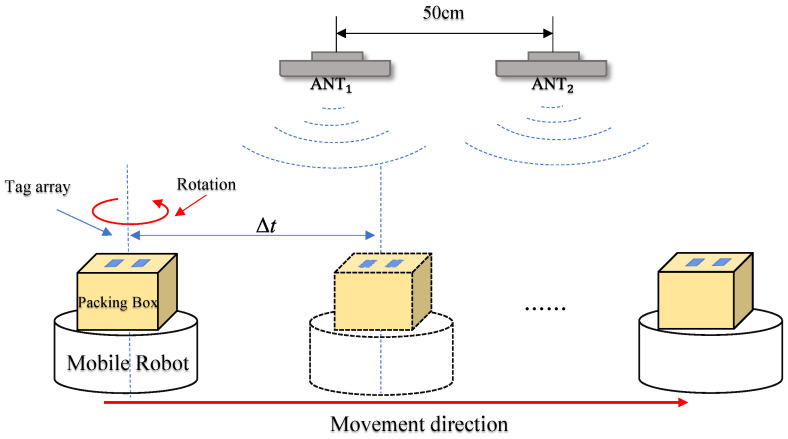
The reader antennas were placed in parallel at 50 cm intervals, the dual-tag arrays were attached to the object surface in parallel, and the mobile robot moved carrying the array of tags.

**Figure 13 sensors-24-06305-f013:**
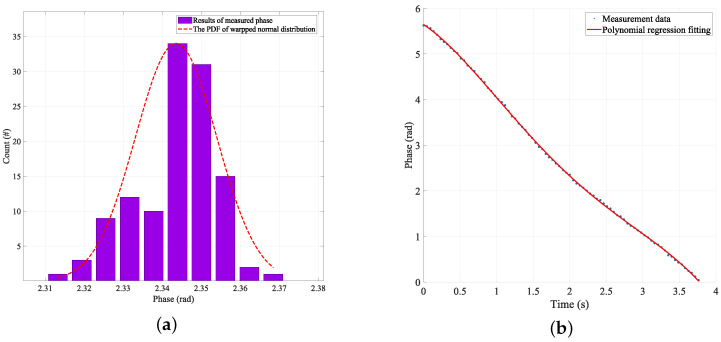
(**a**) Random phase distribution. The tag was fixed at *x* = 100 cm from the reader antenna, and the signal frequency was 920.675 MHz. The measured phase was different every time it was recorded by the reader. (**b**) Polynomial regression of measured phase.

**Figure 14 sensors-24-06305-f014:**
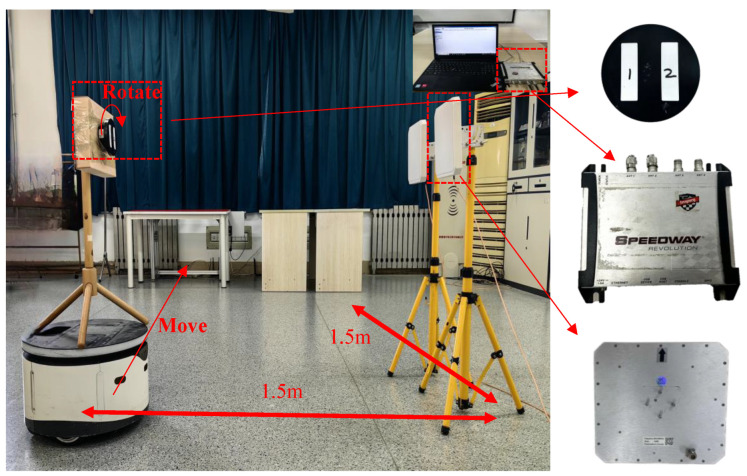
Experiment specific scenario. On the right side of the figure are the reader, the antenna, the rotary table, and the tag array.

**Figure 15 sensors-24-06305-f015:**
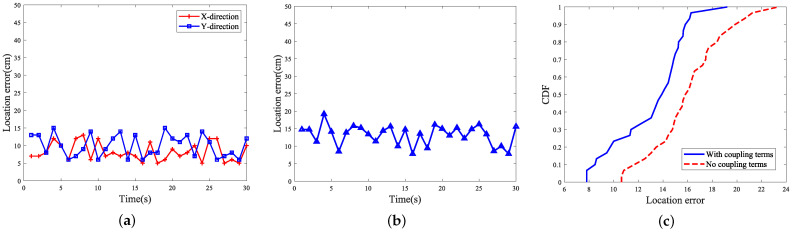
Positioning error for label moving unrotated scenario: where (**a**) shows the positioning error accuracy of the tag in the *x*-axis direction and the *y*-axis direction, (**b**) represents the overall error accuracy in the *x* and *y* directions, and (**c**) denotes the cumulative distribution function of the error with or without considering the coupling in the model.

**Figure 16 sensors-24-06305-f016:**
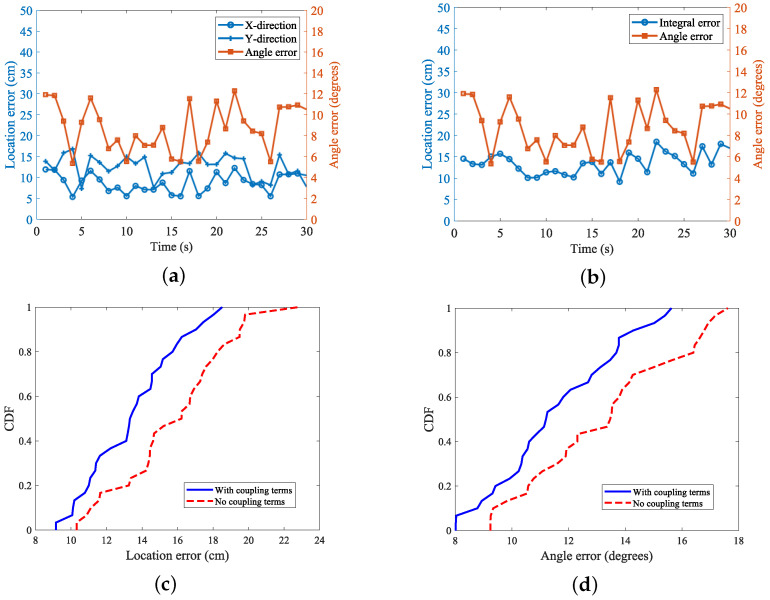
Positioning error in the label moving with rotation scenario: (**a**) *x*- and *y*-axis direction errors; (**b**) overall labeling error; (**c**) positioning with and without the coupling term CDF; (**d**) angle with or without coupling term CDF.

**Figure 17 sensors-24-06305-f017:**
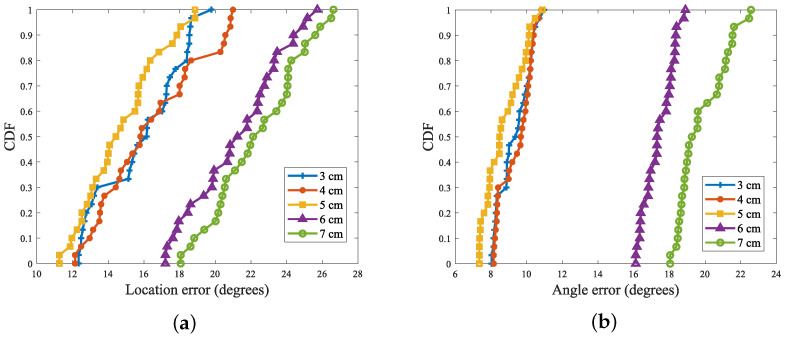
Effect of tag spacing on positioning and angle recognition accuracy. (**a**) The effect of tag spacing on positioning and recognition accuracy; (**b**) The effect of label spacing on angle recognition accuracy.

**Figure 18 sensors-24-06305-f018:**
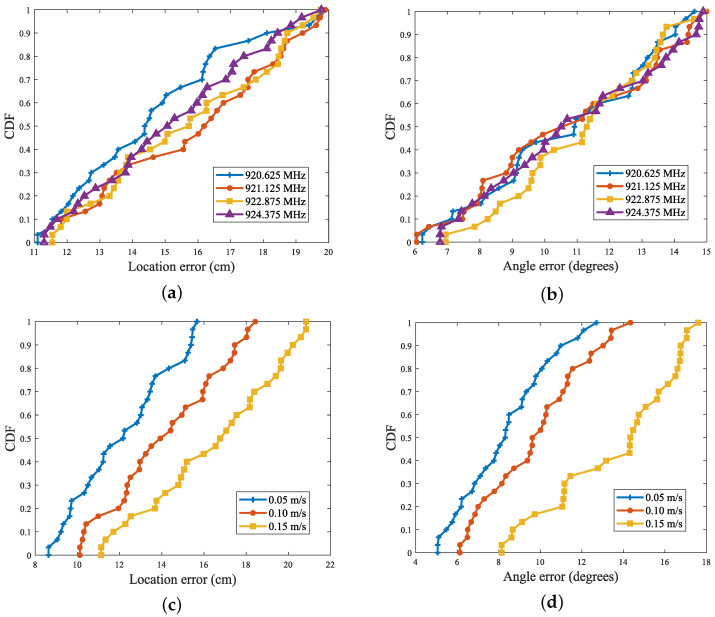
(**a**,**b**) Effect of reader operating frequency on positioning and angle recognition accuracy; (**c**,**d**) Effect of tag movement speed on positioning and angle recognition accuracy.

**Table 1 sensors-24-06305-t001:** Parameter settings.

Parameter	Value
Robot operation speed	0.05m/s
Reader read power	22dBm
Antenna center/Tag height	100cm
Antenna’s location	(0,50) (0,100)

## Data Availability

The data supporting the reported results in this study are not publicly available due to privacy restrictions.

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
