# Peer review of "Tag-Array-Based UHF Passive RFID Tag Attitude Identification of Tracking Methods"

_sensors, 2024, doi:10.3390/s24196305_

Round 1
Reviewer 1 Report
Comments and Suggestions for Authors
I have the read paper carefully, and I assume it needs some work to be done, please refer to the following comments:
· In my view the authors should mention is free test of tag arrangement for identifying the tracking their orientation; no other scattering objects were applied.
· I wish to know the time slots taken when reading tag 1 and then tag 2. This is not mention in the paper.
· How the above time considered with the exact reference of phase angle.
· Sub-title the graphs in figure 1.
· Where is the antenna D in Figure 6?
I assume the results are interested but it needs more attention when scattering objects were tackled with the test measurements.
Comments on the Quality of English LanguagePLease see my comments above.
Reviewer 2 Report
Comments and Suggestions for Authors
I think your manuscript can make a significant contribution to RFID-based object attitude sensing, supported by a solid theoretical foundation and well-executed experiment. It seems that the research design is robust, and the findings are strongly supported by the data. However, I suggest that you make minor amendments in the following areas:
· List the limitations of the approach, especially when it is used in a wider real-world scalable commercial application and suggest some possible future research.
· Briefly address the ethical considerations and potential security vulnerabilities.
· A more in-depth discussion of the results' implications would also improve the overall presentation.
· Simplify some sentences and reduce redundancy throughout the text to enhance readability.
· When referencing equations in the text, ensure that each reference is clear and accompanied by the necessary context. When introducing a new equation, please explain its importance and how it integrates into the overall methodology to enhance understanding. For instance, in equation 1 clarify the practical meaning of the phase components (θT, θR, θTAG) and how each influences the phase measurement. Further, include brief discussion for equation 10, how the rotation formula transforms coordinates would clarify its impact on the tag's position relative to the antenna. Also breaking down equation 11 will minimize the complexity of multiple trigonometric functions to show how the transformation is applied to the coordinates. I think, equation 15 is fundamental to the optimization process can you include small numerical example to demonstrate how the terms interact. For equation 18 you may need to provide additional context to highlight its importance and how that affect the system performance.
· For Fig. 4 use distinct colors to differentiate axes (x, y, z) and include legend directly in the figure, then increase the font size of labels and legends. The same applied to Fig. 5. Fruther consider splitting Fig. 8 into subfigures a, b, and c. and Fig. 13 is a little bit blurred, consider add shading to the figure. For Fig. 15 clearly differentiate between the conditions with and without the coupling term using distinct line styles or colors. Then include error bars or confidence and add detailed caption.
Good luck!
Comments on the Quality of English LanguageIn general, the manuscript is well-written. However, the language could be clearer and simpler in some sections. Hence, minor grammatical edits and improvement in wording could improve readability. In conclusion, although the language is generally proficient, thorough proofreading is recommended.
Reviewer 3 Report
Comments and Suggestions for Authors
Please see the attached PDF

Comments on the Quality of English LanguageSome sentences are lengthy and complex, which can affect readability. Consider breaking longer sentences into shorter, clearer ones.
Round 2
Reviewer 3 Report
Comments and Suggestions for Authors
I don't have any other comments, the author has already addressed the comments in the last round review